# Evaluation of the Anti-Spike (RDB) IgG Titer among Workers Employed at the University of Pisa Vaccinated with Different Types of SARS-CoV-2 Vaccines

**DOI:** 10.3390/vaccines10081244

**Published:** 2022-08-03

**Authors:** Rudy Foddis, Riccardo Marino, Roberto Silvestri, Poupak Fallahi, Salvio Perretta, Christian Garaffa, Riccardo Morganti, Martina Corsi, Jonathan Mennucci, Francesco Porciatti, Gianluca Nerli, Rodolfo Buselli, Antonello Veltri, Fabrizio Caldi, Giovanni Guglielmi, Grazia Luchini, Silvia Briani, Donatella Talini, Francesco Cipriani

**Affiliations:** 1Department of Translational Research and New Technologies in Medicine and Surgery, University of Pisa, 56126 Pisa, Italy; rudy.foddis@unipi.it (R.F.); poupak.fallahi@unipi.it (P.F.); jmennucci.md@gmail.com (J.M.); francescoporciatti93@gmail.com (F.P.); g.nerli90@gmail.com (G.N.); 2Department of Biology, University of Pisa, 56126 Pisa, Italy; roberto.silvestri@biologia.unipi.it; 3Occupational Health Department, Azienda Ospedaliera-Universitaria Pisana, 56126 Pisa, Italy; salvio.perretta@gmail.com (S.P.); dott.martinacorsi@gmail.com (M.C.); r.buselli@ao-pisa.toscana.it (R.B.); antonelloveltri@gmail.com (A.V.); f.caldi@ao-pisa.toscana.it (F.C.); g.guglielmi@ao-pisa.toscana.it (G.G.); 4Faculty of Medicine and Surgery, University of Pisa, 56126 Pisa, Italy; c.garaffa@studenti.unipi.it; 5SOD Statistical Support for Clinical Trials, Azienda Ospedaliera-Universitaria Pisana, 56126 Pisa, Italy; r.morganti@ao-pisa.toscana.it; 6Direzione Aziendale, Azienda Ospedaliera-Universitaria Pisana, 56126 Pisa, Italy; grazia.luchini@ao-pisa.toscana.it (G.L.); silvia.briani@ao-pisa.toscana.it (S.B.); 7CeRIMP—Centro di Riferimento per gli Infortuni e le Malattie Professionali UF PISLL, Dipartimento della Prevenzione ASLNO, 52100 Arezzo, Italy; donatella.talini@uslnordovest.toscana.it; 8UFC Epidemiologia e UFS Cerimp, Dipartimento di Prevenzione Azienda USL Toscana Centro, 52100 Arezzo, Italy; francesco1.cipriani@uslcentro.toscana.it

**Keywords:** SARS-CoV-2 infection, COVID-19, vaccines, healthcare workers, occupational medicine

## Abstract

With the development of SARS-CoV-2 vaccines, many authors started evaluating the immunization efficacy of the available vaccines mainly through sero-positivity tests or by a quantitative assessment of the IgG against the spike protein of SARS-CoV-2 virus in vaccinated subjects. In this work, we compared the titers resulting from vaccination and tried to understand the potential factors affecting the immune response to the available SARS-CoV-2 vaccines. This study was conducted on 670 volunteers employed at the University of Pisa and undergoing a health surveillance program at the University Hospital of Pisa. For each participant, 10 mL of blood, information about contacts with confirmed cases of COVID-19, age, sex, SARS-CoV-2 vaccination status, previous SARS-CoV-2 infection and symptoms, type of vaccine and the date of administration were collected. In the multivariate analysis, the type of vaccine, the presence of symptoms in SARS-CoV-2 positive individuals, and the distance from the second dose significantly affected the antibody titer; the combined vaccination resulted in a faster decay over time compared with the other types of vaccination. No significant differences were observed between Spikevax and Comirnaty (*p* > 0.05), while the antibody levels remain more stable in subjects undergoing Vaxzevria vaccination (*p* < 0.01) compared with mRNA-based ones.

## 1. Introduction

The SARS-CoV-2 outbreak has represented a global health issue affecting not only the general population worldwide but also many workers exposed to higher work-related risks of SARS-CoV-2 infection, for which the implementation of adequate prevention protocols is still challenging. With the development of SARS-CoV-2 vaccines, many authors started evaluating the immunization efficacy of the available vaccines mainly through sero-positivity tests or by a quantitative assessment of the IgG against the spike protein of SARS-CoV-2 virus in vaccinated subjects [1,2]. However, partly due to poor attention to the potential role of factors influencing immune response and anti-spike IgG production in vaccinated individuals, researchers have so far failed to identify protective antibody titer to discriminate between effectively immunized people and people with low immunity and high risk of infection.

In Italy, the vaccination campaign against the SARS-CoV-2 virus began on 31 December 2020. Priority was initially given to health workers and to people affected by chronic diseases with a high impact on health and to older people. These categories had access to m-RNA vaccine, such as Comiranty. Afterwards, some professional categories were identified to have higher priority on vaccination than general population. These included teachers and school staff of all levels. In these categories, Vaxzevria was administered, depending on regional vaccination plan. The University of Pisa has also promoted the diffusion of vaccination among its employers.

In this work, we analyzed the anti-spike antibody titer among vaccinated workers of the University of Pisa enrolled in health surveillance programs. We compared the titers resulting from different vaccination types and tried to understand the potential factors that can have affected the immune response to the available SARS-CoV-2 vaccines.

## 2. Materials and Methods

### 2.1. Study Population

This study was conducted on 670 volunteers employed at the University of Pisa and undergoing a health surveillance program at the University Hospital of Pisa and recruited between May and August 2021.

Subjects were contacted through an email from the University that informed them of the possibility to attend the test and that gave them all the information on the study. They were asked to reserve a blood test around 30 days after their second dose of vaccination. As they came to have their blood sample collected, information about contacts with confirmed cases of COVID-19, age, sex, SARS-CoV-2 vaccination status, type of vaccine and the date of administration of the second dose was collected for each participant. For subjects who developed COVID-19 before completing the vaccination scheme, data about the experienced symptoms and the need for hospitalization were also collected. All subjects who reported SARS-CoV-2 infection had developed the infection before the second dose of the vaccine; consequently, in accordance with Italian law, they had only performed one vaccination dose after natural immunization. Subjects for which vaccination data were unavailable were excluded from the study.

The study was conducted in accordance with the Declaration of Helsinki and approved by the Ethics Committee of Comitato Etico Regionale per la Sperimentazione Clinica della Regione Toscana Sezione: AREA VASTA NORD OVEST, Prot: COV19VACC, 31 January 2022. All participants provided the written informed consent. 

### 2.2. Antibody Titer

For each participant, 10 mL of blood was collected 30 or more days after the second dose (except for the unvaccinated subjects) to evaluate the relationship between the antibody titer and other variables. The titer of the IgG antibodies against the receptor binding domain (RBD) of the SARS-CoV-2 spike protein was determined with a chemoluminescence-based assay on the Abbott Architect platform according to the manufacturer’s instructions. All samples have been handled according to the OMS guideline for manipulating potentially infected biological samples [1,3,4,5]. 

### 2.3. Statistical Analysis

In this prospective study, the sample size was calculated considering a greater ability of Comirnaty vaccine to produce antibodies anti SARS-CoV-2 than the Vaxzevria vaccine. Effect size (difference between Comirnaty and Vaxzevria) and standard deviation were assessed to be around 100 UA/mL and 350 UA/mL, respectively. In this condition setting power at 95% and α-error at 5% it was necessary to enroll 670 subjects.

Categorical data were described by absolute and relative frequency, continuous data by median and interquartile range. To compare “anti SARS-CoV-2 antibody titer” with continuous factors (age, distance from booster dose) Spearman’s correlation analysis was applied. The comparison between “anti SARS-CoV-2 antibody titer” and categorical factors (sex, type of vaccine, SARS-CoV-2 infection, SARS-CoV-2 positivity, symptoms) was performed by Mann–Whitney test or Kruskal–Wallis test followed by multiple comparisons with Bonferroni method. All factors resulting significant to the univariate tests were analyzed together by a multiple linear regression model as multivariate analysis. Regression coefficient with 95% confidence interval, standard error, and beta value were indicated and Durbin–Watson parameter to assess autocorrelation in the residuals due to the model was also calculated. Anti SARS-CoV-2 antibody titer were also compared with symptoms, hospitalization, dyspnea, fever > 38 °C, pneumonia, headache, ageusia/anosmia and cough among the SARS-CoV-2 positive individuals using Mann–Whitney test. Significance was fixed at 5% and all analyzes were carried out with SPSS technology v.27.

## 3. Results

### 3.1. Study Population

The study population included 670 subjects, of which 279 (41.6%) were male and 391 (58.4%) females, with a median age of 50 (IQR 41–57). Among these, three were not vaccinated, 493, 142, and 16 were vaccinated with two doses of Vaxzevria, Comirnaty, or Spikevax, respectively. For 16 subjects that received the first dose of Vaxzevria and a second dose of an mRNA-based vaccine (e.g., Comirnaty or Spikevax), the vaccination type has been reported as “combined”. Nineteen participants (2.8%) developed COVID-19 before completing the vaccination scheme, thus receiving only one dose (Table 1). 

Among the confirmed COVID-19 cases, 89.5% developed disease-related symptoms, with fever and cough being the most frequent (47.4 and 63.2%, respectively). Only three patients (15.8%) required hospitalization.

### 3.2. Factors Affecting the Antibody Levels in Vaccinated People

The univariate analysis reported that the type of vaccine significantly affects the antibody titer (*p* < 0.001). Similarly, a confirmed infection of SARS-CoV-2 and the development of COVID-19 symptoms were associated with a higher antibody level (*p* = 0.002 and 0.047, respectively). A positive correlation was observed between the antibody titer and the distance from the second dose (*p* = 0.045); no correlation with age was observed (*p* > 0.05) (Table 2 and Table 3). 

To evaluate how different types of vaccines affected the titer, we carried out a Bonferroni corrected pairwise comparison. Vaccination with Spikevax or a combination of Vaxzevria and an mRNA-based vaccine led to the highest antibody levels, with a median IgG AU of 12,127.5 and 7935.7, respectively. Two doses of Comirnaty resulted in significantly higher antibody levels than two doses of Vaxzevria (IgG AU 2933 and 1141.9, respectively; *p* < 0.0001). The detailed results of this comparison are reported in Table 4.

In the multivariate analysis, the type of vaccine, the presence of symptoms in SARS-CoV-2 positive individuals (but not the SARS-CoV-2 infection), and the distance from the second dose significantly affected the antibody titer, as shown in Table 5. 

Interestingly, a negative correlation emerged between the titer and the distance from the second dose.

When we evaluated the relationship between the antibody titer and the distance from second dose in different types of vaccine, we found that the combined vaccination resulted in a faster decay over time compared with the other types of vaccination. No significant differences were observed between Spikevax and Comirnaty (*p* > 0.05), while the antibody levels remain more stable in subjects undergoing Vaxzevria vaccination (*p* < 0.01) compared with mRNA-based vaccines (Figure 1).

## 4. Discussion

In this work, we reported that factors such as the type of vaccination, the development of a symptomatic SARS-CoV-2 infection and the distance from the second dose could affect the levels of the anti-spike antibodies among people vaccinated against COVID-19. Distance from second dose showed different behaviors in uni- and multivariate analysis. It is possible that the time between vaccine and testing differed according to the type of vaccine administered; correcting for the type of vaccine, the distance showed a negative correlation with the antibody titer, as would be expected. Interestingly, the occurrence of interstitial pneumonia or the need for hospitalization were associated with a higher, although not statistically significant, antibody titer. These observations are consistent with the findings of other groups showing a higher antibody titer in subjects with a symptomatic SARS-CoV-2 infection [6,7,8,9,10] and in those patients who developed a severe form of COVID-19 [11,12,13].

We did not observe any significant correlation between the antibody titer and gender or age. Although this observation was consistent with the finding of other studies [14,15], others reported significantly higher antibody levels among young people and females undergoing SARS-CoV-2 vaccination [16,17,18,19].

In previous works, Dorschug et al. [14] and Peterfhoff et al. [20] reported higher antibody titers among subjects receiving a single dose of Comirnaty compared with subjects receiving a single dose of Vaxzevria. Our study reinforced this observation by showing that mRNA-based vaccinations resulted in higher antibody levels compared with Vaxzevria, even after the second dose. Interestingly, a combination of Vaxzevria (as a first dose) and Comiranty (as a second dose) seems to result in higher levels of anti-spike antibodies and neutralizing IgG compared with a complete cycle of Vaxzevria or Comirnaty [21,22]. Consistently, in our cohort, the subjects who received a combined vaccination (Vaxzevria + an mRNA-based vaccine) showed higher antibody titers than those who received the same type of vaccine as the first and second dose. However, despite the initial higher titer, the combination of Vaxzevria with an mRNA-based vaccine led to a faster decay over time of the IgG anti-spike levels compared with a complete cycle of Comirnaty or Spikevax, with Vaxzevria generating the most stable response. Interestingly, a similar decay was observed by other authors in subjects vaccinated with Comirnaty [23,24].

While these observations suggest the importance of the booster dose, they do not allow establishing an antibody cut-off level to estimate an eventual loss of protection from SARS-CoV-2 infection over time. In a study conducted on Rhesus macaques, McMahan et al. reported that the adoptive transfer of purified IgG from convalescent to naïve macaques protected these latter against SARS-CoV-2 challenge in a dose-dependent manner. The authors showed that even low antibody titers were sufficient for protection against SARS-CoV-2, with cellular immune response contributing to protection when antibody responses were suboptimal [22]. Additionally, in a clinical trial conducted to evaluate the correlation between neutralizing antibody levels and COVID-19 risk, Gilbert et al. found that the risk of developing COVID-19 was negatively correlated with the antibody titers after Spikevax administration [25]. Overall, these observations may explain how even the low titers achieved with Vaxzevria after the second dose may have contributed to reducing the COVID-19 risk in vaccinated subjects. 

This work has several limitations. Firstly, the serological tests were not followed by neutralization assays, which can make the interpretation of the results a bit challenging, although some data suggest a correlation between the levels of the total and neutralizing IgG [23]. Additionally, some asymptomatic SARS-CoV-2 infections in our cohort may have gone undetected. Moreover, we lack information about comorbidities or therapy that can influence antibodies titer. Lastly, given the reduced number of subjects vaccinated with Spikevax, vaccinated with a combination of Vaxzevria + an mRNA-based vaccine, or with confirmed infection of SARS-CoV-2, further analysis on a larger cohort may be needed to confirm our results.

## 5. Conclusions

Although some of these data are present in the literature, our study highlighted that the type of vaccine, a SARS-CoV-2 infection, and the symptom’s severity are the most relevant factors affecting the antibody titer among people vaccinated against COVID-19. Conversely, we did not find any correlation between the titer and the age or sex. Moreover, Vaxzevria showed a lower but more stable response.

The data reported in this work may contribute to highlighting the factors affecting the immune response in vaccinated subjects, however no conclusion can be drawn about the efficacy in protecting against SARS-CoV-2 infection or reducing the severity of the symptoms.

Despite the fact our study represents only a small and partial step in this direction, identifying factors affecting the immune response would be crucial in the evaluation and organization of vaccination campaigns aimed at the general population, addressing the choices of vaccine schedule and of the categories of population at risk to be prioritized.

## Figures and Tables

**Figure 1 vaccines-10-01244-f001:**
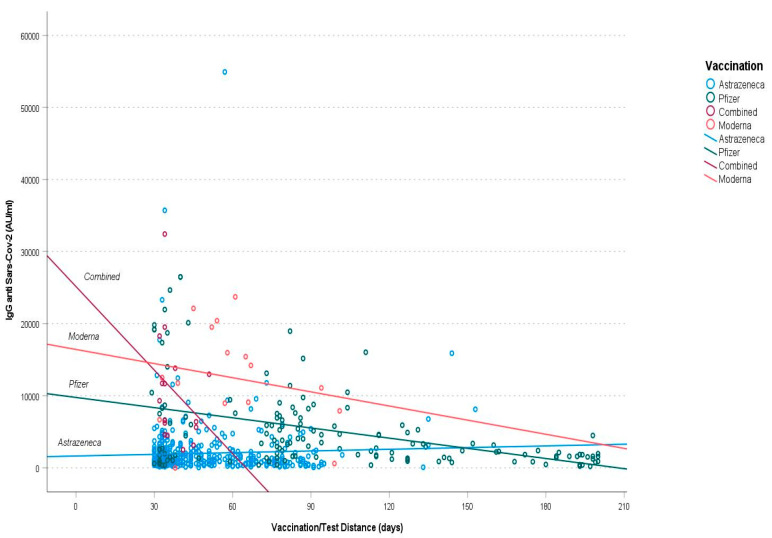
Antibody titer decreasing over time. Purple represents Combined Vaccination, red Moderna, green Pfizer and blue Astrazeneca.

**Table 1 vaccines-10-01244-t001:** Study population characteristics. Statistics: frequency (%) or median (interquartile range).

Factor	Statistics
**Age (years)**	50 (41–57)
**Sex**	
M	279 (41.6)
F	391 (58.4)
**Virus SARS-CoV-2 IgG_BAU**	205.2 (106.3–454.9)
**Virus SARS-CoV-2 IgG AU**	1444.8 (748.7–3203.5)
**Distance from booster dose (days)**	42 (34–74)
**Vaccine type**	
**Unvaccinated**	3 (0.4)
**Vaxzevria (Astrazeneca)**	493 (73.6)
**Comirnaty (Pfizer)**	142 (21.2)
**Spikevax (Moderna)**	16 (2.4)
**Combined (Vaxzevria + Comirnaty or Spikevax)**	16 (2.4)
**Vaccinated**	667 (99.6)
**Contact with a SARS-CoV-2-positive subject**	76 (11.3)
**Confirmed SARS-CoV-2 infection**	19 (2.8)

**Table 2 vaccines-10-01244-t002:** Univariate analysis of the factors influencing “anti SARS-CoV-2 antibody titer”, expressed in UA/mL.

Factor	Univariate Analysis	
** *Continuous Factors* **	**Spearman’s rho**	** *p* ** **-value**
**Age**	0.046	0.239
**Distance from booster dose**	0.078	0.045
** *Categorical Factors* **	**Median (IQR)**	** *p* ** **-value**
**Sex**		0.624
M	1435.7 (759.1–2728.5)	
F	1454.0 (748.4–3319.4)	
**Type of vaccine**		<0.001
(1) Unvaccinated	1287.1 (820.5–1988.0)	
(2) Vaxzevria	1141.9 (642.0–2208.2)	
(3) Comirnaty	2933.0 (1438.2–6704.2)	
(4) Combined	7935.7 (4854.0–13,602.5)	
(5) Spikevax	12,127.5 (8167.8–18,624.8)	
**SARS-CoV-2 infection**		0.002
No	1390.8 (742.3–3132.2)	
Yes	3003.0 (1759.8–8685.2)	
**SARS-CoV-2 positivity**		0.06
Unvaccinated	1287.1 (820.5–1988.0)	
Vaccinated	5656.6 (2302.4–11,006.1)	
**Symtoms**		<0.001
No	1373 (737–2869)	
Yes	6336 (2691–11,780)	

**Table 3 vaccines-10-01244-t003:** Comparison between “anti SARS-CoV-2 antibody titer”, expressed in UA/mL and several symptoms among the SARS-CoV-2 positive individuals.

Factor	Median (IQR)	*p*-Value
**Symptoms**		0.047
No	591.5 (413.6–769.4)	
Yes	5415.6 (2410.0–10,232.0)	
**Hospitalization**		0.254
No	2846.9 (1405.2–8207.5)	
Yes	8122.6 (5405.7–31,526.0)	
**Dyspnoea**		0.536
No	2846.9 (1405.2–8207.5)	
Yes	5415.6 (2625.1–26,463.8)	
**Fever > 38 °C**		0.604
No	2658.0 (680.4–13,862.7)	
Yes	5415.6 (2224.3–7729.9)	
**Pneumonia**		0.109
No	2846.9 (1405.3–7785.6)	
Yes	26,463.8 (14,576.3–40,696.6)	
**Headache**		0.701
No	2688.9 (1028.2–14,398.3)	
Yes	4450.3 (2458.1–8025.9)	
**Ageusia/Anosmia**		1.000
No	2690.9 (1482.1–14,398.3)	
Yes	5656.6 (1641.6–8025.9)	
**Cough**		0.592
No	2690.9 (769.4–8685.2)	
Yes	4209.3 (2302.4–17,114.2)	

**Table 4 vaccines-10-01244-t004:** Multiple comparisons by Bonferroni method for the antibody titer among different types of vaccine.

Vaccine	Median (IQR)	Vaccine	Median (IQR)	*p*-Value
Vaxzevria	1141.9 (642.0–2208.2)	Comiranty	2933.0 (1438.2–6704.2)	<0.001
		Spikevax	12,127.5 (8167.8–18,624.8)	<0.001
		Combined	7935.7 (4854.0–13,602.5)	<0.001
Comiranty	2933.0 (1438.2–6704.2)	Spikevax	12,127.5 (8167.8–18,624.8)	0.018
		Combined	7935.7 (4854.0–13,602.5)	0.003
Spikevax	12,127.5 (8167.8–18,624.8)	Combined	7935.7 (4854.0–13,602.5)	0.653

**Table 5 vaccines-10-01244-t005:** Multivariate analysis of the factors influencing “anti SARS-CoV-2 antibody titer”. Vaccine types have been numbered from 1 to 5 based on the median antibody titer (Table 4). RC: regression coefficient; SE: standard error; Beta: beta coefficient. Durbin–Watson parameter is equal to 1.9 indicating absence of autocorrelation in the residuals due to the model.

Factor	RC	95% CI	SE	Beta	*p*-Value
**Vaccination**	4020	3507; 4534	261	0.525	<0.001
**Distance from booster dose**	−28	−37; −19	4.6	−0.210	<0.001
**SARS-CoV-2 infection**	−1053	−6776; 4669	2914	−0.033	0.718
**COVID-19 symptoms**	9914	3806; 16,024	3111	0.288	0.002
** *Constant of the model* **	−4775	−5961; −3589	604		<0.001

## Data Availability

The data can be requested from the corresponding author.

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
