# Peer review of "Evaluation of the Anti-Spike (RDB) IgG Titer among Workers Employed at the University of Pisa Vaccinated with Different Types of SARS-CoV-2 Vaccines"

_vaccines, 2022, doi:10.3390/vaccines10081244_

Round 1
Reviewer 1 Report
introduction: it introduces the topic
methods are well described
Results: I think you should modify the table layout
discussion: the discussion develops the problems and the results of the study rightly; by the way, some sentences are heavy to read (e.g.137-142). I think you should discuss better the impact of your research
overall judgment : the paper is well done, but in some parts, it is not easy to understand
Author Response
Dear Reviewer,
We appreciate your interest in our work. As you have pointed out, some sentences in the paper are hard to read. I revised english and the structure of sentences all long the manuscript, especially the discussion part.
We also tried to make the conclusion more understandable.
All the changes from the previous manuscript are marked in yellow.
We further thank you for your time and ask you to reassess the quality of the manuscript
Dr. Riccardo Marino and colleagues

Reviewer 2 Report
The Foddis et al article explor the relationship between IgG level and variables such as type of vaccine, sex, or COVID invection19. It is shown that the type of vaccination, symptomatology and time since the second dose affect IgG-anti-S levels.
However, these data have already been evaluated in the literature and the novelty of this article is not clear to me. On the other hand, it is also not clear what objectives are intended to be achieved when you finish reading the introduction.
I also consider that the protocol of section 2.1 should be deeply explained. And, the statistical section, how the data are described (mean, median, se, IQR...) and how the regression models were constructed. I also suggest that the article be reviewed by a statistician. Why are the estimated beta coefficients +/- SE not shown in table 3?
Minor comments:
1. Write the reference of the ethics committee and the institution that endorsed it in the text.
2. Line 78 replace distance by time.
3. Line 89, give some kind of deviation.
4. Exclude the category "no" from the tables as it is overunderstood and describe the table footnote.
Author Response
Dear Reviewer,
We appreciate your interest in our work. We have taken into consideration the major and minor issues you have pointed out. We’ll reply one by one:
- “However, these data have already been evaluated in the literature and the novelty of this article is not clear to me. On the other hand, it is also not clear what objectives are intended to be achieved when you finish reading the introduction”,
We are aware that our paper is just one of many studies about COVID-19 vaccine. However, we would like to give our effort to the subjects. We tried to explain better the aim of the study in inte introduction and make it more understandable.
- “I also consider that the protocol of section 2.1 should be deeply explained. And, the statistical section, how the data are described (mean, median, se, IQR...) and how the regression models were constructed. I also suggest that the article be reviewed by a statistician. Why are the estimated beta coefficients +/- SE not shown in table 3?”
We have our work revised a second time from a statistician. The “Statistical analysis” section was modified in ordet to answer to the request and furthermore beta coefficient and standard error were also indicated (new table V).
Minor comments:
- Write the reference of the ethics committee and the institution that endorsed it in the text.
We have add it in the Materials and Method section.
- Line 78 replace distance by time.
We have replaced it.
- Line 89, give some kind of deviation.
We have done it, using IQR.
- Exclude the category "no" from the tables as it is overunderstood and describe the table footnote.
We have removed the no category aside from where it is usefull to show the difference between the two categories (as in Table VI).
All the changes from the previous manuscript are marked in yellow.
We further thank you for your time and ask you to reassess the quality of the manuscript
Dr. Riccardo Marino and colleagues

Reviewer 3 Report
I was invited to revise the paper entitled "Evaluation of the anti-spike (RDB) IgG titer among workers employed at the University of Pisa vaccinated with different types of SARS-CoV-2 vaccines".
It was a cross-sectional study aimed to evaluate differences in IGG titer by covid-19 vaccine schedule.
Despite the topic is relevant, this paper presents several limitation:
- Introduction section is too poor. Authors did not presented the vaccination campaign and how vaccination was promoted in Italy;
- Sample size estimation was totally missing;
- The enrollment procedure was not reported;
- A COVID infection occurred prior the vaccination or between the two doses can influence the titer response. Authors have to explain the period of infection patients with prior COVID-19;
- Data reported in table 2 were useless. The aim of this study was not to report covid-19 symptoms;
- Authors stated that IGG titers were not normally distributed so linear regression analysis was not appropriate;
- A strong limitation of this study was the difference among study groups. Combined and Spikevax groups are too small;
- Authors should analyze titers by age groups. Older patients were vaccinateds prior to younger, according to the national vaccination plan and it can results in different distance to vaccination by age group. In addition, the lack of informations about comorbidities is an important limitation. older patients or patients with immune system diseases can develop a lower IGG titer.
Author Response
Dear Reviewer,
We appreciate your interest in our work. We have taken into consideration the major and minor issues you have pointed out. We’ll reply one by one:
- “It was a cross-sectional study aimed to evaluate differences in IGG titer by covid-19 vaccine schedule.”
The study does not aim to evaluated just the difference by vaccine schedule, but also othe variables affecting antybody titer. We have tried to explain it better in the revised manuscript
- “Introduction section is too poor. Authors did not presented the vaccination campaign and how vaccination was promoted in Italy”.
You were right. We enriched Introduction section and write down a paragraph about vaccination campaing in Italy.
- “The enrollment procedure was not reported”
We reported the procedure in the Matherials and Methods section
- “A COVID infection occurred prior the vaccination or between the two doses can influence the titer response. Authors have to explain the period of infection patients with prior COVID-19”
All the COVID19 infection that we have found happened before the vaccination campaign. According to Italian law, people with infection had just one dose given. So, in our study, both vaccinated with no infection and with infection have two “doses”. We’ve reported this on the manuscript too.
- “Data reported in table 2 were useless. The aim of this study was not to report covid-19 symptoms”
Data in Table II just describes symptoms characteristics. This is because, after that, we shoed difference of IGG titer according to some variables, including symptoms.
- “Sample size estimation was totally missing;
- Authors stated that IGG titers were not normally distributed so linear regression analysis was not appropriate;
- A strong limitation of this study was the difference among study groups. Combined and Spikevax groups are too small”
The sample size calculation was indicated in the new “Statistical analysis” section. The distribution of the dependent variable was not very different from the normal distribution and so a multiple linear regression was applied and supported by Durbin-Watson (DW) parameter calculation to assess autocorrelation in the residuals due to the model. If the DW parameter is around 2 the autocorrelation is absent and this parameter was found to be equal to 1.9 indicating absence af autocorrelation. Combined and Spikevax groups are too small but these two vaccines are the ones administered far less even in the general population.
- “Authors should analyze titers by age groups. Older patients were vaccinateds prior to younger, according to the national vaccination plan and it can results in different distance to vaccination by age group. In addition, the lack of informations about comorbidities is an important limitation. older patients or patients with immune system diseases can develop a lower IGG titer.”
In our study, we haven’y found any correlation between age and IGG titer. So we did not analyze it based on age groups. But it is true that one of the limitations of our papaer is the lack of information about comorbidities. We have added to the limitations of the study.
All the changes from the previous manuscript are marked in yellow.
We further thank you for your time and ask you to reassess the quality of the manuscript
Dr. Riccardo Marino and colleagues

Round 2
Reviewer 2 Report
I wanted to thank the authors for this revised version. The objectives have been made more concrete. If a statistician revises the article, I have no more concretizations.
Author Response
Dear Reviewer,
We are glad you appreciate our paper and the efforts to improve it.
Greeting,
Dr. Riccardo Marino and colleagues

Reviewer 3 Report
Authors tried to address all comments raised during the first round of revisions. Despite it, I remain skeptical about several points:
- About sample size estimation, Authors stated (lines 94-98) "the sample size was calculated considering a greater ability of Comirnaty vaccine to produce antiboides anti Sars-Cov-2 than the Vaxzevria vaccine.Effect size (difference between Comirnaty and Vaxzevria) and standard deviation were assessed to be around 100 UA/ml and 350 UA/ml". It is unknown where Authors find this information, no references were cited;
- Despite the explanation, table 2 remain useless. It should be removed because it reported redundant results. Authors can maintain the table 6 that report same data.;
- About age groups, The absence of correlation does not mean that age di not impact the IGG titer. Subgroup analysis should be performed;
The last main observation is about the clinical relevance of these findings: despite significative differences, all vaccination strategies showed strong IGG respond. So, what is the clinical significance of these findings?
Author Response
Dear Reviewer,
We’d like to improve our paper thanks to you suggestions. We have taken into consideration the new issues you have pointed out. We’ll try to reply one by one:
1) “About sample size estimation, Authors stated (lines 94-98) "the sample size was calculated considering a greater ability of Comirnaty vaccine to produce antiboides anti Sars-Cov-2 than the Vaxzevria vaccine.Effect size (difference between Comirnaty and Vaxzevria) and standard deviation were assessed to be around 100 UA/ml and 350 UA/ml". It is unknown where Authors find this information, no references were cited”.
The median of Astrazeneca in our study is about 1000 AU / ml and therefore a variation of 100 AU / ml represents 10% as occurs in many superiority studies like ours (generally the range for superiority goes from 10% to 20% and is at the discretion of the Authors).
2) “Despite the explanation, table 2 remain useless. It should be removed because it reported redundant results. Authors can maintain the table 6 that report same data”.
We accepted the suggestion of the reviewer keeping only table VI.
3) “About age groups, The absence of correlation does not mean that age di not impact the IGG titer. Subgroup analysis should be performed”.
Comparing the antibody titers with the age classes through the Kruskal-Wallis test we obtain a p-value equal to 0.161
These are the values (AU / ml ) of the medians and the IQR relating to the various age groups:
<30y 1460 (815-2690)
30-40y 1111 (682-2393)
40-50y 1494 (769-3488)
50-60y 1329 (634-2395)
>60y 1634 (780-4765)
If you think it could be useful, we can add this analysis and results in a table.
4) “The last main observation is about the clinical relevance of these findings: despite significative differences, all vaccination strategies showed strong IGG respond. So, what is the clinical significance of these findings?”.
Our work aims to evaluate the antibody titer and the factors that influence it but, as specified in the discussion and in the conclusion sections, it does not allow clinical evaluations on the disease output. The added value to the literature of this work is to confirm the different antigenity of m-RNA vaccines compared to viral vector vaccines and to show that the antibody titer of the latter ones remains stable longer over time, albeit lower. Furthermore, it would seem to suggest that age is not a factor influencing the antibody titre, although this data is however limited by the size of our sample. Identifying factors affecting the immune response would be crucial in the evaluation and organization of vaccination campaigns aimed at the general population, addressing the choices of vaccine schedule and of the categories of population at risk to be prioritized. We’ve changes part of the conclusion in order to make these aspects more clear.
All the changes from the previous manuscript are tracked.
We further thank you for your time and ask you to reassess the quality of the manuscript
Dr. Riccardo Marino and colleagues
